# Nature Makes No Leaps: Building Continuous Location Embeddings with Satellite Imagery from the Web

## Abstract

Building location embedding from web-sourced satellite imagery has emerged as an enduring research focus in web mining. However, most existing methods are inherently constrained by their reliance on discrete, sparse sampling strategies, failing to capture the essential spatial continuity of geographic spaces. Moreover, the presence of confounding factors in satellite images can distort the perception of actual objects, leading to semantic discontinuity in the embeddings. In this work, we propose SatCLE, a novel framework for Continuous Location Embeddings leveraging Satellite imagery. Specifically, to address the out-of-distribution query challenge of spatial continuity, we propose a geospatial refinement strategy comprising stochastic perturbation continuity expansion and graph propagation fusion, which transforms discrete geospatial coordinates into a continuous space. To mitigate the effects of confounders on semantic continuity, we introduce causal refinement, integrating causal theory to localize and eliminate spurious correlations arising from the environmental context. Through extensive experiments, SatCLE shows state-of-the-art performance, exhibiting superior spatial coherence and semantic fidelity across diverse geospatial tasks. The source code is available at https://anonymous.4open.science/r/SatCLE.

## Keywords

Location Embedding, Satellite Imagery, Continuity, Web Mining

## 1 Introduction

With the ever-growing availability of geospatial data on the web, the effective representation and understanding of location have become critical research priorities. *Location embedding*, a prominent and enduring theme within the domain of web mining and knowledge discovery, involves encoding geographical locations across the globe as dense vectors in a latent space, as depicted in Figure 1. By integrating both spatial relationships and contextual nuances from diverse web sources (*e.g.*, satellite [25, 34] and street-view imagery [58]), location embeddings significantly enhance the capacity to analyze and predict spatial patterns, thereby improving the accuracy and efficiency of decision-making across a wide range of applications [73], including web computing [32, 62], urban planning [20, 67], and location-based social networks [12, 15, 68].

Compared to web-sourced data such as geo-tagged images [36, 58, 69], points of interests (POI) [33, 69] and road networks [26], *satellite imagery* has emerged as a mainstream modality for learning location embeddings [73]. Its widespread accessibility and ability to provide rich semantic information – ranging from natural environments and urban layouts to road networks – have contributed to its growing adoption in web research. This trend is also driven by the recent advances in deep learning methods [40, 70], which have proven highly effective in extracting and learning visual features.

Existing studies [25, 34] on learning location embeddings with satellite imagery primarily centers on a fundamental principle:

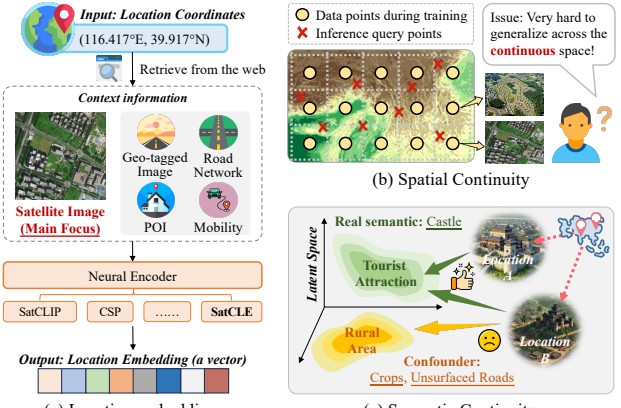

**Figure 1: Location embedding and its challenges.**

the significance of geographic coordinates arises not from their numerical values but from the contextual information associated with the geographical entities they represent, encompassing social, environmental, and geographical aspects. Early attempts mostly concentrate on the use of explicit or manual location encodings (*e.g.*, one-hot mapping [69], information theory [66], sinusoidal functions [33]), which typically require intricate designs and exhibit limited generalization capabilities across different domains. To mitigate this issue, *Contrastive Learning* (CL) has gained traction within the web research community [63], aiming to develop more flexible and general-purpose location embeddings, such as CSP [34] and SatCLIP [25]. The key insight is to learn implicit location embedding by matching extracted visual patterns of web-sourced satellite imagery with their geographic coordinates. Further, the resulting location encoder can efficiently summarize the characteristics of any given location for convenient use in downstream tasks.

Though promising, these CL-based methods overlook another fundamental principle of location embeddings – **Continuity**. As Leibniz famously stated, *"Nature makes no leaps"*, a principle derived from classical philosophy and science, suggesting that natural processes unfold gradually, without sudden or abrupt changes. In the context of location embedding, this implies that geographical and environmental changes occur incrementally, in continuous steps rather than large, discontinuous leaps. In this paper, we argue that the concept of continuity in location embeddings inherently manifests from two distinct perspectives:

◆ **Spatial Continuity.** The overhead perspective of satellite imagery offers an exceptional means for capturing geographical context. Although there are already global satellite platforms [46, 61] available, the balance between spatial resolution and data volume remains a persistent trade-off in learning location embedding from global-level satellite imagery. Obtaining appropriate semantic context for locations [17, 25] demands kilometer-level resolution, which necessitates datasets on the scale of tens of millions. Therefore, the current mainstream datasets for satellite image-based location

embedding [25] adopt global sparse sampling, where points are randomly and non-adjacently sampled, resulting in inherent distances between them. As illustrated in Figure 1 (b), due to the lack of training data, capturing continuous spatial features near intermediate points (marked with a red cross) poses a challenge. In particular, without significantly increasing the dataset size, this spatial sampling sparsity leads to out-of-distribution (OoD) query problems, causing generalization errors during inference. To achieve more fine-grained semantic modeling, it is crucial to capture the spatial continuity inherent in the geographical data.

✦ **Semantic Continuity**. In addition to spatial continuity, another scenario emerges where data points, although spatially distant, display notable similarity. Upon projection into the latent space, these points similarly exhibit a form of continuity, which we refer to as semantic continuity. As verified in previous studies [59, 72], satellite images often contain *confounding factors* that can distort the perception of actual objects. This issue also arises when learning location embeddings from satellite imagery, and can lead to semantic discontinuity. For instance, Figure 1 (c) depicts two structurally similar castles located in distant regions. Location A, a castle situated in a well-developed tourist area in China, is accurately represented. However, Location B, despite sharing similar architectural features, is located in a remote area of India, surrounded by dense crops and unsurfaced roads. Such environmental context (*e.g.*, crops) led to the misunderstanding of Location B as a farm, acting as a confounding factor in this scenario. In the presence of confounding factors, existing methods [25, 34] in location embedding frequently struggle to maintain semantic continuity, as models often project semantically similar elements into distinct latent spaces. Therefore, it is essential to develop an embedding approach that can effectively mitigate confounding influences, thereby addressing semantic continuity while enhancing both the precision and coherence of semantic representations.

In this paper, we present SatCLE, a continuous location embedding framework with satellite Imagery. To address the first challenge, we design a geospatial refinement process consists of two part: stochastic perturbation continuity expansion and graph propagation fusion. Stochastic perturbation continuity expansion incorporates random perturbations to transform discrete geospatial coordinates into a continuous space. Subsequently, the graph propagation fusion is employed to integrate the features of permuted locations, effectively mitigating the spatial OoD issues that arise from insufficient coverage. To address the second challenge, we introduce a causal refinement approach that leverages causal theory to identify semantically meaningful patches and mitigate interference from environmental confounders through a backdoor adjustment mechanism. Our method achieves state-of-the-art performance compared with other baselines. Specifically, it achieves an average improvement of 15.03% in mean squared error for regression tasks and an average improvement of 7.7% in accuracy for classification tasks.

In summary, our contributions lie in the following aspects:

- *Geospatial refinement for spatial continuity*. We first propose a geospatial refinement strategy that utilizes stochastic perturbation continuity expansion and graph propagation fusion to integrate random perturbations, transforming discrete geospatial coordinates into a continuous space to preserve spatial continuity.

- *Causal refinement for semantic continuity*. We also develop a causal refinement strategy, which is the first to apply causal theory to location embedding. This strategy employs a semantic attention mechanism to identify semantically meaningful patches and utilizes backdoor adjustment to mitigate spurious correlations resulting from environmental confounding factors.

- *Extensive empirical studies*. Our SatCLE framework demonstrates superior performance across a range of tasks, enhancing generalization and robustness in location embeddings.

## 2 Preliminary

### 2.1 Formulation

**Definition 1 (Geospatial Location)** . It is identified using latitude and longitude, which is a spherical coordinate system utilizing the surface of a sphere in three dimensions to define spatial coordinates on Earth. This system is capable of precisely pinpointing any location on the Earth. Each geospatial location can be denoted as $(\lambda_i, \theta_i)$, where $\lambda_i \in [-\pi, \pi]$ and $\theta_i \in [-\pi/2, \pi/2]$ .

**Definition 2 (Satellite Image)**. A satellite image provides a comprehensive view of a geographical area from a top-down perspective, offering rich information. For a satellite image, where the center point corresponds to a given geospatial location, it is denoted as: $I \in \mathbb{R}^{\mathcal{H} \times \mathcal{W} \times 3}$, where $\mathcal{H}$ and $\mathcal{W}$ are height and width.

**Definition 3 (Geospatial Indicator)**. A geospatial indicator serves as a reflection of environmental or socio-economic conditions (*e.g.*, elevation, population, carbon emission) on Earth. It enables predictions based solely on latitude and longitude coordinates, providing a means to evaluate whether a model has effectively learned semantically rich location embeddings. The $M$ indicators on a set of $K$ locations on earth are denoted as $Y \in \mathbb{R}^{M \times K}$.

**Problem Statement (Location Embedding)**. Given the coordinate $L$ with its associated satellite imagery $I$, the main goal is to learn the location embedding $h$ and accurately estimate the socioeconomic indicator $y$. The process can be formulated as: $h = \mathcal{F}(I, L)$, $\mathcal{F}$ is a mapping function learned during training.

### 2.2 Related Work

#### 2.2.1 Location Embedding from Web-Sourced Data.
With the widespread adoption of sensor-equipped smartphones, a vast amount of data uploaded to the Internet from around the world is increasingly associated with GPS coordinates, thereby giving rise to the task of location embedding and its wide range of applications. [8, 33, 38, 53, 69]. Loc2Vec [53] first queries neighbouring points in a GIS database for a given location and visualizes these points as images using Mapnik [5], thereby learning location embeddings directly from the semantic context. GPS2Vec+ [69] segments the Earth according to the Universal Transverse Mercator (UTM) grid, representing these divisions using one-hot encoding.

Given the wide accessibility and the unique overhead perspective provided by satellite imagery, the current state-of-the-art research [25, 34] prioritizes satellite imagery as the preferred supplementary modality for location embedding. CSP [34] introduces a dual-encoder architecture designed to separately encode both images and their corresponding locations. SatCLIP [25] leverages spherical harmonics as positional encoders, complemented by sinusoidal representation networks, enhancing its ability to model the

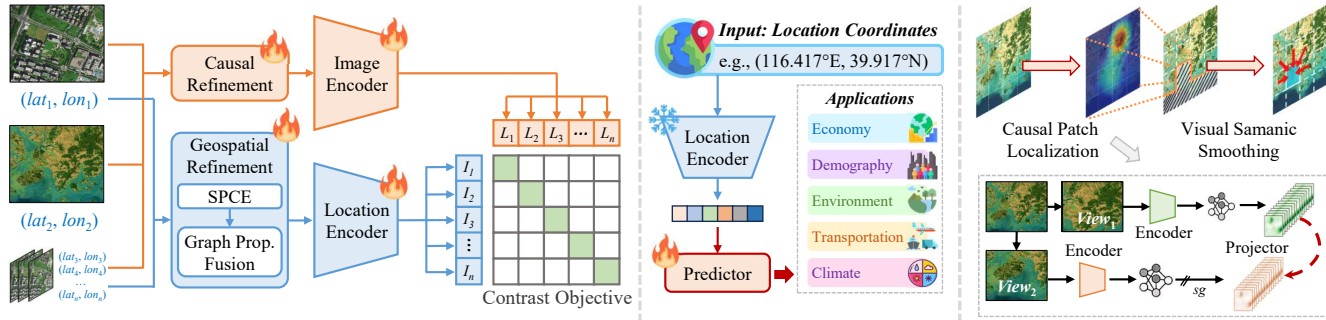

(a) Phase 1: Pretraining  (b) Phase 2: Downstream Tasks  (c) Causal Refinement

**Figure 2: The overall framework of our proposed SatCLE. *sg* denotes stop gradient.**

spatial complexity inherent in satellite vision. However, previous works overlooked the continuity inherent in the physical world, which resulted in the emergence of OoD bias. *To address this, we propose a novel continuous location embedding framework, integrating Geospatial Refinement and Causal Refinement to better capture the seamless nature of real-world spatial data from the web.*

*2.2.2 Geospatial Contrastive Pretraining with Satellite Imagery.* Geospatial Learning with Satellite Imagery faces the challenge of using vast amounts of unlabeled satellite data due to the high cost and expertise required for labeling [20, 63, 65, 67]. As a result, Contrastive learning has increasingly evolved into a fundamental paradigm for effectively leveraging the data [60]. After comprehensive literature review, we summarize two main approaches in geospatial contrastive pretraining are: (a) Vision Pretraining, which uses satellite imagery to represent geographic locations, reducing the need for ground-based surveys. Methods like Tile2Vec [23] and SeCo [37] enhance image learning but face limitations in accuracy due to the absence of explicit location information. (b) Vision-Location Pretraining, which integrates location coordinates with visual data. SatCLIP [25], GeoCLIP [58], and Sphere2Vec [36] are notable methods in this category. *Remarkably, our work follows the approach of Vision-Location Pretraining, highlighting the long-overlooked importance of continuity and bridging the gap from both semantic and spatial continuity perspectives.*

*2.2.3 Causal Inference.* Causal inference [11, 30, 59, 64] explores how changes in certain variables impact outcomes, offering researchers a powerful method to assess the true relationship between variables. Recent work integrates deep learning with causal methods, extending its application to computer vision [11, 28]. For instance, [11] enhances zero-shot learning by applying counterfactual interventions to establish substantial visual-semantic correlations, resulting in more robust visual classification. Additionally, only a limited number of studies have extended this approach to spatial data mining [30, 54, 64]. For example, [30] employs backdoor adjustment to mitigate confounding factors in trajectory sequences, while [59] utilizes attention mechanisms to identify causal features, improving model robustness across different spatio-temporal tasks. Nevertheless, to the best of our knowledge, there is little work connecting causal inference to location embedding tasks. *Consequently, we adopt a causal perspective for the first time to investigate confounding factors in location embedding and use causal techniques to refine the semantic continuity in geographical visual contexts.*

## 3 Methodology

Figure 2 depicts the dual-phase framework of our SatCLE :

- **Phase 1**: We initially obtain an image-location pair as input, where the location refers to the coordinates of the central point of each satellite image. The satellite image and location are processed through their respective unimodal encoders independently. While extracting unimodal features, the location undergoes geospatial refinement, and the satellite image is subject to causal refinement, both aimed at enhancing the continuity of their respective features. Subsequently, a contrastive interaction mechanism is developed to align the representations of these two modalities within the latent space.
- **Phase 2**: In the downstream task prediction phase, we get rid of the vision encoder and only utilize a frozen location encoder for predicting downstream tasks, by simply finetuning multi-layer perceptrons with a few trainable parameters.

### 3.1 Modality Representation Learning

We leverage Vision Transformer (ViT) [19] as the visual encoder to process satellite imagery, which consists of alternating layers that perform the multi-head self-attention (MSA) operation [57] and fully-connected (FC) layers. Additionally, layer normalization (LN) is applied before each block, while residual connections are implemented following each block. For the location modality, we propose using spherical harmonic basis functions as positional embeddings [49], providing effective global coverage of geospatial data. More details on modality learning can be found in Appendix B.

### 3.2 Geospatial Refinement for Spatial Continuity

Spatial continuity is a key aspect of continuous location embedding. The current challenge lies in the construction of the dataset of satellite image-geocoordinate pairs [25, 34], which is derived from the random sampling of publicly available satellite imagery with global coverage. Therefore, each sampled point is inherently associated with a specific spatial distance from others, as illustrated in Figure 11 in the Appendix. However, during inference, the input data can correspond to any arbitrary location on Earth, resulting in an out-of-distribution spatial shift, *i.e.*, $P(X_{te}) \neq P(X_{tr})$. From a learning perspective, while the principle of empirical risk minimization is effective, considering functions (*i.e.*, neural networks) even with a parameter count comparable to the training sample size may lead to simple memorization, making it difficult for the

model to generalize beyond specific locations seen during training. Thus, to achieve smoother uncertainty estimation, another effective principle is vicinal risk minimization [71]. In practice, we enhance geospatial refinement through spatially continuous augmentation, thereby improving model robustness in downstream tasks.

Next, we introduce our spatial continuity augmentation method, which consists of two key components: Stochastic Perturbation Continuity Expansion (SPCE) to achieve a continuous probability distribution and Graph Propagation Fusion for feature integration.

*3.2.1 Stochastic Perturbation Continuity Expansion.* As discussed above, the geospatial coordinates corresponding to each satellite image can be regarded as a discrete sample, with finite distances between adjacent points. To tackle this challenge, we first propose the SPCE mechanism. By introducing random perturbations, we can transform discrete geospatial coordinates into a continuous space. Specifically, given a set of discrete coordinates $L = \{l_1, l_2, ..., l_n\}$, where each coordinate $l_i \in \mathbb{R}^2$ represents a location on the Earth's surface. Without augmentation, these coordinates are discrete, with no overlap between different points $l_i$ and $l_j$. Then we introduce a random perturbation to each point, resulting in the new shifted coordinate $l_i' = l_i + \Delta l_i$, where $\Delta l_i$ is a random variable sampled from the distribution $p(\Delta l_i)$, which could be a Gaussian distribution $\mathcal{N}(0, \sigma^2)$ or another type of distribution. This operation creates a continuous range of perturbations around each point.

*3.2.2 Graph Propagation Fusion.* After the SPCE module introduces perturbations, the augmented coordinates are no longer isolated but form a continuous region centered around $l_i$. This effectively expands the surrounding space of the discrete points, gradually connecting the originally scattered point set $\{l_1, l_2, ..., l_n\}$ into a geometrically continuous space. To propagate spatial relationships between different location features, we employ a Graph Neural Network (GNN) for feature fusion. In line with Tobler's First Law of Geography [39], which states that nearby objects are more likely to share similar characteristics, we construct an adjacency matrix $A$ based on the distance between coordinates:

$$A_{ij} = exp(-\frac{d_{ij}^2}{2\sigma^2}), \ d_{ij} = ||l_i' - l_j'||, \tag{1}$$

where $\sigma$ denotes the standard deviation. Through the GNN, the location feature $z_i^l$ is aggregated with its neighboring features, producing the fused feature:

$$h_i^{k+1} = \phi(\sum_{j \in \mathcal{N}} A_{ij} W^k h_j^k), \tag{2}$$

where $h_i^{k+1}$ denotes the feature representation of node $i$ at layer $k$, $W_k$ is the learnable weight matrix, $\phi$ is the non-linear activation function. $\mathcal{N}$ is the neighborhood set generated by SPCE of node $i$.

*3.2.3 Discussion.* The essence of data augmentation lies in altering the distribution of the original points, so that we no longer rely solely on discrete coordinate points. Instead, by adding noise, these points become more broadly distributed across the space. The augmented coordinates $l_i'$ follow the distribution:

$$p(l_i') = \int p(l_i) \, p(\Delta l_i) \, d(\Delta l_i), \tag{3}$$

where $p(l_i')$ represents the distribution of the original discrete coordinates, and $p(\Delta l_i)$ is the distribution of perturbations. This formula shows that the perturbed point $l_i'$ is no longer a single discrete point but forms a continuous Gaussian distribution around $l_i'$. As the perturbation $\sigma$ increases, the spatial gaps between geocoordinates are gradually filled, thus achieving continuity. This process can be described in terms of continuity using the following inequality:

$$\lim_{\sigma \to 0} p(l_i') = \delta(l - l_i). \tag{4}$$

As the perturbation $\sigma$ approaches zero, $p(l_i')$ degenerates into the original discrete distribution $\delta(l - l_i)$, which is the Dirac delta function representing an isolated point. Conversely, when $\sigma$ is sufficiently large, $p(l_i')$ gradually covers the region surrounding $l_i$, exhibiting continuity. Through graph propagation fusion, the model further smooths these location features in the latent space, ensuring continuity in both geometric and latent space.

## 3.3 Causal Refinement for Semantic Continuity

The presence of confounding factors is a primary challenge to maintaining semantic continuity. Therefore, an effective location embedding method should be able to effectively mitigate confounding influences, ensuring that data with similar semantic patches are accurately projected to proximate locations within the semantic space. In this part, we first adopt a causal inference [45] perspective to provide a theoretical foundation for understanding the impact of confounding factors on the final semantic representations (*e.g.*, the existence of backdoor paths). Subsequently, we introduce backdoor adjustment techniques to mitigate the effects of these confounding factors. Finally, we present the practical implementation process.

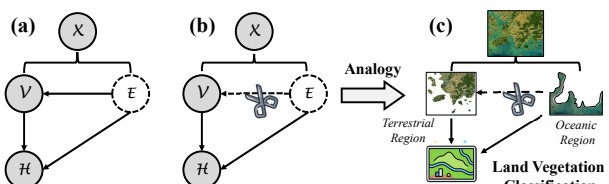

**Figure 3: SCM for Causal Patch Localization.**

*3.3.1 A Causal Look on Semantic Continuity.* Formally, we carefully examine the existing process of satellite visual representation learning and construct a Structural Causal Model (SCM) to analyze the causal relationships in location embeddings, as shown in Figure 3 (a). The SCM consists of three components: visual semantic representation $V$, environmental semantic representation $E$, and the final representation $H$ used for location embedding, with arrows indicating causal relations. In the context of satellite image inputs, $V$ and $E$ are complementary and mutually exclusive, together forming the complete input $X$. The actual image components corresponding to each part can be found in Figure 3 (c). In the SCM, beyond the intuitive visual encoding process from $V \to H$ and $E \to H$, the transition from $E \to V$ further illustrates that the representation of visual information is strongly influenced by environmental factors. The castle example discussed in Section 1 is an intuitive case.

Therefore, this establishes a backdoor path as $V \leftarrow E \to H$. Here, environmental information serves as a confounding factor influencing both the visual semantic data $X$ and the visual representation $H$. Accordingly, we propose the utlization of causal intervention

techniques on variable $V$ to address the confounding influences of variable $E$ (Figure 3 (b)), thereby enhancing the learning of robust representations from inherently biased visual data.

*3.3.2 Backdoor Adjustment.* Building on the previous causal analysis, our method for learning visual semantic representations focuses on eliminating backdoor paths instead of directly modeling the association $P(H \mid V)$. Grounded in causal theory [44], we utilize the robust technique of backdoor adjustment, which enables us to obstruct the backdoor path by estimating $P(H \mid do(V))$, where $do(\cdot)$ denotes the *do-calculus*.

$$
\begin{aligned}
P(H \mid do(V)) &= \sum_i^\eta P(H \mid do(V), e_i)\, P(e_i \mid do(V)) \\
&= \sum_i^\eta P(H \mid do(V), e_i)\, P(e_i) = \sum_i^\eta P(H \mid V, e_i)\, P(e_i),
\end{aligned}
\tag{5}
$$

where $e_i \in E$ denotes the patches containing environmental information. First, we can re-express $P(H \mid do(V))$ using Bayes' theorem. Next, given that the variables $V$ is not the descendant of $H$, it follows that $P(e_i \mid do(V)) = P(e_i)$. At the same time, since the reaction of $H$ to $V$ and $e_i$ does not affect the causal relationship between $H$ and $V$, we can set the conditional probabilities $P(H \mid do(V), e_i) = P(H \mid V, e_i)$.

*3.3.3 Implementation.* In practical implementation, the process can be divided into two steps: Causal Patch Localization and Attentional Semantic Smoothing. As illustrated in [59], semantically relevant information remains prominent even after undergoing unsupervised training techniques such as view changes and data augmentation. This notion is analogous to the concept of our semantic causal patches (*e.g.*, the terrestrial region in Figure 3 (c)). Consequently, we leverage the pretrained visual attention mechanism [9] during pretraining to distinguish causal content. Its pertaining process is illustrated in Figure 2 (c). The overall structure is a self-distillation framework that maintains a student network (green) and a teacher network (orange), both sharing the same architecture but having different parameters. We apply two distinct affine transformations to an input image, one each for the student and teacher networks. The output of the teacher network is centered using a mean computed over the batch. A stop-gradient (sg) operation is applied to the teacher to ensure that gradients are propagated only through the student. The teacher's parameters are updated using an exponential moving average (ema) of the student's parameters, and their similarity is measured through cross-entropy loss.

After localizing the causal patches, it is imperative to mitigate the influence of environmental factors and extraneous variables in order to maintain semantic consistency throughout the pretraining process. We perform the attentional semantic smoothing on the non-causal patches, *i.e.*, replacing them with semantic blocks from the causal patches, which can be formulated as:

$$
P'_i = \begin{cases} P_i, & \text{if } A_i \geq \tau \text{ (causal patches)} \\ \frac{1}{|\mathcal{S}(P_i)|} \sum_{P_j \in \mathcal{S}(P_i)} P_j, & \text{if } A_i < \tau \text{ (non-causal patches)}, \end{cases}
\tag{6}
$$

where $A_i$ is the attention value of each patch, $\tau$ is the threshold for identifying a causal patch, $P_i / P'_i$ is the original/new value of i-th patch, and $\mathcal{S}(P_i)$ denote the nearby patches of $P_i$.

## 3.4 Pretraining & Finetuning

*3.4.1 Pretraining Stage.* The overall goal of our framework involves optimizing the contrastive loss:

$$
\mathcal{L}_{\text{contrast}} = -\log \frac{\exp\left(\text{sim}(h_i, v_i)/\tau\right)}{\sum_k \exp\left(\text{sim}(h_i, v_k)/\tau\right)},
\tag{7}
$$

where $h_i$ and $v_i$ denote the normalized embedding of location representation and image representation in the $i$−th pair respectively. In this equation, $\alpha$ and $\beta$ represent weight hyperparameters. By employing back propagation optimization, we align multi-granularity cross-modal urban image-text input data, leading to the development of a robust encoder.

*3.4.2 Finetuning Stage.* After the pretaining phase, we can obtain the final satellite image-enhanced location embedding $h$. During the finetuning stage, as illustrated in Figure 2 (b), we keep the location embedding $h$ frozen and a Multi-Layer Perception (MLP) classifier is trained on top to finetune the prediction of geo spatial indicators, represented as $Y = \text{MLP}(h)$.

## 4 Experiments

In this section, we evaluate our method to answer the following research questions (RQs):

- **RQ1:** Does SatCLE surpasses existing methodologies in performance and demonstrate robust generalization across diverse socioeconomic and environmental applications?
- **RQ2:** What are the individual contributions of the various components of SatCLE to its overall effectiveness?
- **RQ3:** To what extent does SatCLE exhibit transferability across different urban environments?
- **RQ4:** How does SatCLE perform qualitatively in real-world case studies, and what insights can be drawn from its predictions when applied to diverse urban environments?

## 4.1 Experimental Setup

*4.1.1 Datasets.* We utilize the open-source S2-100k dataset [25] as our pretraining dataset, which is derived from Sentinel-2 satellite imagery. This dataset consists of 12 channels and spans all seven continents, offering the most extensive and balanced spatial coverage compared to other popular datasets, such as [14, 34, 55].

The S2-100k dataset encompasses imagery from 2021 to 2023, formatted into 256x256 pixel image patches in GeoTIFF format, with each pixel representing a resolution of 10 meters. Each satellite image is associated with the corresponding latitude and longitude of the patch's center point, utilizing the EPSG:4326 coordinate system. Follow [25], we randomly select 90% of the data points for pretraining, ensuring uniform distribution, while the remaining 10% is set aside as a validation set to check for overfitting.

*4.1.2 Downstream Tasks.* In our work, the Downstream Dataset is pivotal in assessing the practical implications of our SatCLE framework. We gather five representative socio-economic and environmental indicators: *Population, Elevation, Carbon Emissions, Country Code* and *Land Vegetation*. Each of these indicators sheds light on critical aspects of earth-based observations. For all the downstream dataset, we split the downstream dataset randomly into training, validation, and testing sets with a ratio of 7:1:2.

**Table 1: Performance of different models on various tasks. The best results are in bold, and the second-best results are underlined. The last row indicates the relative improvement in percentage.**

| Dataset | Global | | | | | | | | North America | | | | | | | |
|---|---|---|---|---|---|---|---|---|---|---|---|---|---|---|---|---|
| Model | Population | | Elevation | | Carbon | | Country | Land Veg. | Population | | Elevation | | Carbon | | Country | Land Veg. |
| | MSE↓ | MAE↓ | MSE↓ | MAE↓ | MSE↓ | MAE↓ | Acc↑ | Acc↑ | MSE↓ | MAE↓ | MSE↓ | MAE↓ | MSE↓ | MAE↓ | Acc↑ | Acc↑ |
| OneHot | 1.144 | 0.749 | 0.673 | 0.517 | 0.712 | 0.632 | 0.715 | 0.254 | 3.767 | 1.330 | 0.882 | 0.534 | 1.893 | 0.837 | 0.214 | 0.098 |
| Sinusoid | 0.455 | 0.470 | 0.524 | 0.495 | 0.702 | 0.601 | 0.843 | 0.102 | 2.690 | 1.103 | 0.586 | 0.488 | 1.785 | 0.823 | 0.353 | 0.268 |
| CSP (iNat) | 0.231 | 0.292 | 0.230 | 0.296 | 0.642 | 0.572 | 0.962 | 0.560 | 0.690 | 0.654 | 0.337 | 0.380 | 1.275 | 0.692 | 0.407 | 0.484 |
| CSP (FMoW) | 0.338 | 0.370 | 0.424 | 0.426 | 0.687 | 0.587 | 0.874 | 0.510 | 0.729 | 0.657 | 0.554 | 0.516 | 1.288 | 0.704 | 0.402 | 0.398 |
| GeoCLIP | 0.377 | 0.354 | 0.186 | 0.259 | 0.565 | 0.552 | 0.967 | 0.513 | 0.716 | 0.520 | 0.423 | 0.428 | 2.165 | 0.608 | 0.227 | 0.364 |
| SatCLIP | 0.204 | 0.275 | 0.120 | 0.206 | 0.568 | 0.559 | 0.952 | 0.525 | 0.679 | 0.593 | 0.240 | 0.331 | 1.162 | 0.617 | 0.562 | 0.497 |
| SatCLE | 0.168 | 0.245 | 0.093 | 0.185 | 0.537 | 0.518 | 0.981 | 0.590 | 0.626 | 0.512 | 0.217 | 0.303 | 1.138 | 0.583 | 0.588 | 0.504 |
| Improvement | 17.65% | 10.91% | 22.50% | 10.19% | 5.21% | 6.56% | 1.43% | 12.38% | 7.81% | 1.56% | 9.58% | 8.46% | 2.07% | 5.51% | 4.63% | 1.41% |

| Dataset | Africa | | | | | | | | Oceania | | | | | | | |
|---|---|---|---|---|---|---|---|---|---|---|---|---|---|---|---|---|
| Model | Population | | Elevation | | Carbon | | Country | Land Veg. | Population | | Elevation | | Carbon | | Country | Land Veg. |
| | MSE↓ | MAE↓ | MSE↓ | MAE↓ | MSE↓ | MAE↓ | Acc↑ | Acc↑ | MSE↓ | MAE↓ | MSE↓ | MAE↓ | MSE↓ | MAE↓ | Acc↑ | Acc↑ |
| OneHot | 5.898 | 1.900 | 1.373 | 0.686 | 3.289 | 1.178 | 0.089 | 0.184 | 0.964 | 0.688 | 0.594 | 0.721 | 3.446 | 1.135 | 0.143 | 0.071 |
| Sinusoid | 3.839 | 1.547 | 1.055 | 0.813 | 3.243 | 1.176 | 0.117 | 0.104 | 0.648 | 0.655 | 0.325 | 0.509 | 3.512 | 4.141 | 0.191 | 0.093 |
| CSP (iNat) | 0.849 | 0.713 | 0.538 | 0.556 | 3.173 | 1.160 | 0.342 | 0.411 | 0.267 | 0.290 | 0.251 | 0.359 | 3.173 | 0.993 | 0.330 | 0.359 |
| CSP (FMoW) | 0.941 | 0.820 | 0.570 | 0.583 | 3.175 | 1.162 | 0.302 | 0.384 | 0.546 | 0.628 | 0.252 | 0.370 | 3.179 | 1.055 | 0.318 | 0.312 |
| GeoCLIP | 1.109 | 0.707 | 0.465 | 0.464 | 3.127 | 1.165 | 0.116 | 0.390 | 0.300 | 0.294 | 0.236 | 0.341 | 3.424 | 1.127 | 0.396 | 0.257 |
| SatCLIP | 0.769 | 0.644 | 0.479 | 0.493 | 3.067 | 1.152 | 0.367 | 0.454 | 0.258 | 0.271 | 0.223 | 0.330 | 2.397 | 1.051 | 0.523 | 0.360 |
| SatCLE | 0.686 | 0.578 | 0.337 | 0.425 | 2.379 | 1.062 | 0.402 | 0.472 | 0.197 | 0.259 | 0.184 | 0.285 | 2.314 | 0.994 | 0.632 | 0.378 |
| Improvement | 10.79% | 10.25% | 37.98% | 8.41% | 28.92% | 7.81% | 9.54% | 3.96% | 23.64% | 4.43% | 17.49% | 13.64% | 3.46% | 5.42% | 20.84% | 5.00% |

- **Population:**. The population density data are subsampled from the global datasets from [48]. The dataset includes a random sample of 10,000 data points from a global scale.
- **Elevation:** Similar to the population density, the elevation figures are subsampled from the global datasets provided by [48]. This dataset comprises a random sample of 10,000 data points on a global scale. The unit is people per sq. km.
- **Carbon Emissions:** Sourced from the Open-sourced Data Inventory for Anthropogenic CO2 (ODIAC) 2022 [42], the dataset includes a random sample of 10,000 data points from a global scale, with emissions quantified in tons on a monthly basis.
- **Country Code:** Following [25], we obtain country boundaries from the 4.1 release of The Database of Global Administrative Areas[1]. The dataset includes a random sample of 10,000 data points from a global scale.
- **Land Vegetation:** The Land Cover data is sourced from [2], which categorizes land cover into 38 distinct classes, such as *cropland_rainfed* and *shrubland*. The dataset includes a random sample of 10,000 data points on a global scale in 2022.

*4.1.3 Baselines.* Intuitively, vision encoders [18, 21] can be readily applied for geospatial indicator prediction by retrieving satellite images corresponding to specific locations. However, due to the limits imposed by free API quota restrictions [1, 4], we were unable to conduct extensive experiments or include comparisons with relevant baselines. Therefore, we compare SatCLE with the following state-of-the-art methods in satellite-based location embedding:

- **OneHot** [14, 69]. The approach utilizes UTM Zones as grid cells to encode geotags globally. However, the number of cells available for encoding geotags is invariably constrained by computational cost and memory limitations.
- **Sinusoid** [33]. This method employs sinusoidal functions to model both absolute locations and spatial contexts, capturing the inherent periodicity and cyclic patterns associated with locations.

---
[1]https://gadm.org/

- **CSP** [34]. A dual-encoder framework encodes images and geographic locations from iNat2018 [56] and FMoW datasets [14], with various pretraining loss options available [8, 23, 27, 37]. Given the similarity in the contrastive framework, training the CSP model on the S2-100K dataset would yield results analogous to SatCLIP. Therefore, follow the established practices [25], we employed CSP pretrained on the FMoW and iNat datasets, hereinafter referred to as CSP(iNat) and CSP(FMoW), respectively.
- **GeoCLIP** [58]. The pioneering work in utilizing GPS encoding for geolocation introduces a novel geo-tagged image-to-GPS location contrastive method. Given the similarity in the contrastive framework, training the GeoCLIP model on the S2-100K dataset would yield results analogous to SatCLIP. We repurpose the model weights trained on the MP-16 dataset [58] due to its extensive global coverage, thereby capable of furnishing a degree of performance reference.
- **SatCLIP** [25]. It introduced CLIP into satellite images, a globally applicable geolocation encoder that learns implicit representations of locations from globally sampled Sentinel-2 satellite data.

*4.1.4 Metrics and Implementation.* To evaluate predictive accuracy, we utilize three widely recognized metrics [25, 34]: mean squared error (MSE), and mean absolute error (MAE) [30, 82] and accuracy (Acc). Better performance is indicated by a lower values for MSE and MAE and a higher Accuracy. Parameter initialization is consistent with the approach described in [25]. During parameter learning, the Adam optimizer is selected to minimize training loss.

A grid search is performed on hyperparameters, with learning rates and batch sizes explored within the ranges {2e-6, 1e-5, 2e-4, 1e-4, 1e-3, 1e-2} and {64, 128, 256, 512, 1024}, respectively. We ultimately selected a batch size of 512 and a learning rate of 1e-4 with 1e-2 wright decay, which produced the best results. We train models for 500 epochs with an early stopping strategy on A100 GPUs with PyTorch 1.13.1 framework on Ubuntu 22.04. We set $K$ as 5, $r$ as 0.01 in Section 3. For $\tau$ in 3.3.3 we set its value to the threshold of the lowest 10%.

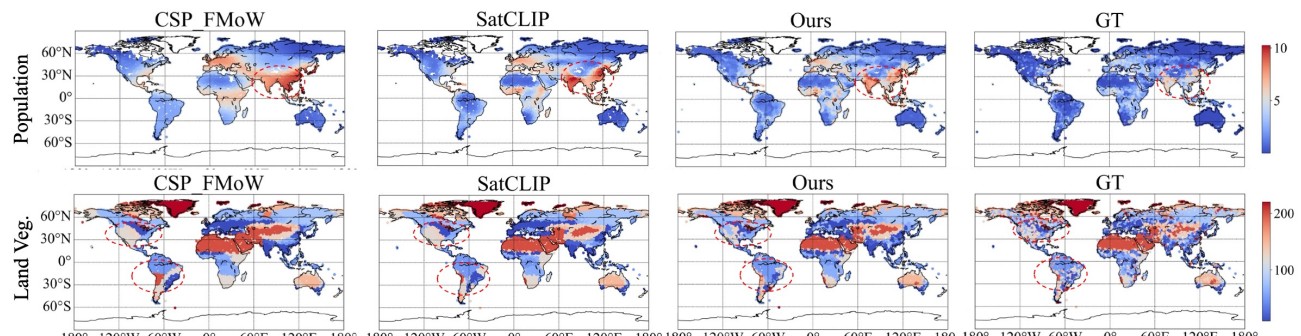

**Figure 4: Population prediction of different models.**

## 4.2 Overall Performance (RQ1)

Table 1 illustrates the overall results on model comparison. As we can see, our model significantly surpasses the baselines, achieving state-of-the-art performance in five downstream tasks, both in regression and classification. The performance gain is 17.6%, 22.5%, 5% in terms of MSE of regression tasks and 3.0%, 12.4% in terms of Accuracy of classification tasks.

In downstream tasks, we can observe that models generally exhibit strong performance in population and elevation prediction in regression tasks, while their performance in carbon estimation may be slightly inferior. This discrepancy could be attributed to the close relationship between carbon emissions and economic activities, with satellite imagery not capturing these factors significantly. For classification tasks, the prediction of country code performs well, possibly owing to that regions within the same country share the same label, thereby avoiding abrupt differences and simplifying the prediction task. In contrast, land vegetation classification is associated with various factors such as human activities, climate types, and topography, resulting in lower prediction accuracy.

## 4.3 Ablation Study (RQ2)

To verify the effects of different components in our `SatCLE` model, we conduct ablation study of our proposed `SatCLE` model and report the results in terms of MSE and Accuracy in Figure 5. As illustrated, the Population and Carbon prediction are likely to be substantially affected by geographic location, which results in a stronger reliance on spatial continuity. In contrast, the performance of other tasks exhibits a greater dependence on semantic continuity. While the significance of spatial and semantic continuity varies across different tasks, the model integrating both forms (`SatCLE`) consistently demonstrates superior performance. This suggests a

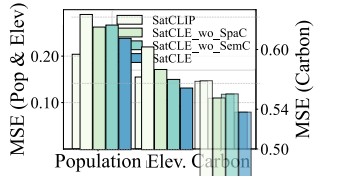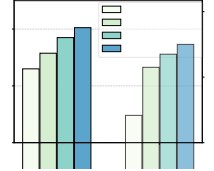

**Figure 5: Ablation study on the core components.** *SpaC*: spatial continuity; *SemC*: semantic continuity.

complementary relationship between these two types of continuity, collectively enhancing the model's ability to capture the spatial and semantic attributes inherent in the data.

## 4.4 Transferability Study (RQ3)

Due to significant differences in the extent of data collection across various regions of the world, researchers in different regions face varying challenges. For instance, in developed cities such as New York [7], Singapore [3], and Hong Kong [6], governments actively promote digitalization and have made a wealth of valuable urban data publicly available. In contrast, researchers working in underdeveloped countries or regions may need to independently collect and annotate the required data. Therefore, we further investigate whether geographic models can learn robust and transferable features within the regions they are trained on, thus helping to alleviate the challenges posed by sparse data label scenarios.

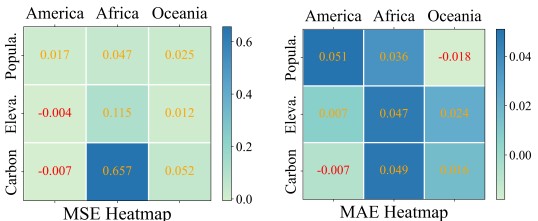

**Figure 6: Transferring capability study.**

Specifically, we evaluate the geographic adaptability of different methods by segmenting the world into regions based on continents. During evaluation, we exclude data from the target continent in the pre-training phase. For downstream tasks, we introduce a small subset of future data from the test continent (randomly selecting 1% uniformly) into the training set, constructing a practical few-shot geographic adaptation scenario. The detailed data distribution for the few-shot scenario is shown in Figure 9 in the appendix.

As seen in Table 1, geographic shifts lead to varying degrees of performance decline. Specifically, across three regression tasks, the average performance on three continents decreased by 0.335, 0.153, and 1.407 in terms of MSE compared to North America and Africa, the performance drop observed in Oceania is less significant, likely due to Oceania's smaller geographic area and relatively uniform environmental conditions [50], reducing the complexity of domain

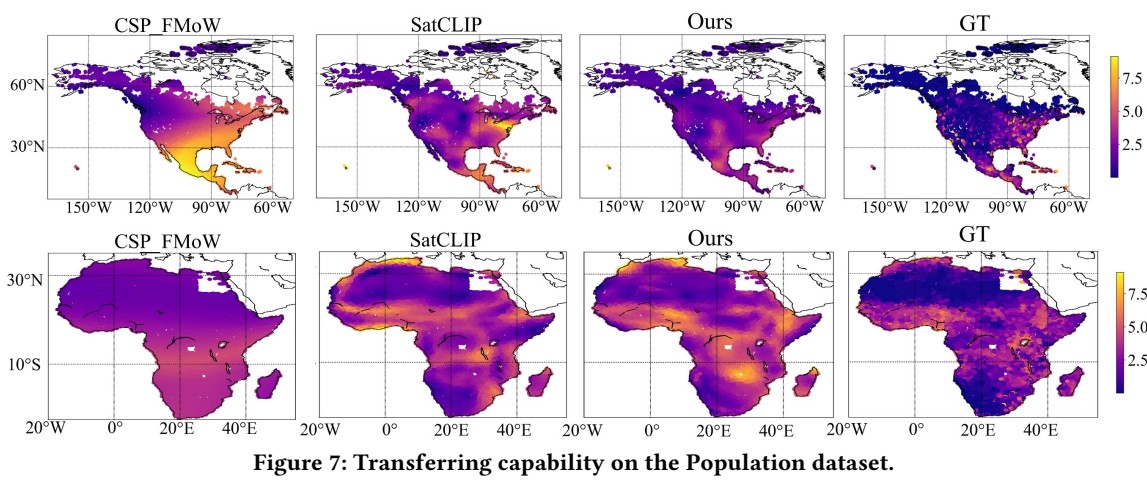

**Figure 7: Transferring capability on the Population dataset.**

adaptation. Moreover, as shown in Figure 7, which plots the performance difference between SatCLE and SatCLIP across continents, nearly all yellow indicators highlight the consistent superiority of SatCLE's transferability across different continents and tasks.

## 4.5 Qualitative Analysis (RQ4)

*4.5.1 Visualization of predicted results.* To effectively illustrate our model's performance, we visualize the predicted outcomes across two distinct indicators: population and land vegetation. As demonstrated in Figure 4, compared with SatCLIP, our SatCLE model incorporates a continuity mechanism that effectively corrects certain inaccurate predictions, particularly in addressing the extreme values within population forecasting. while CSP demonstrates overall over-smoothed predictions, showing minimal abrupt value fluctuations across different locations.

*4.5.2 Visualization of geographic adaptation results.* We also visualize the transferability results on population forecasts in North America and Africa in Figure 7. As observed, CSP, due to its reliance on classical self-supervised methods and issues such as the sparsity of the training dataset and cross-domain challenges, exhibits overly smooth continuous transitions. On the other hand, SatCLIP shows abrupt and unnatural changes in spatial representations, which may be attributed to overfitting during training. In contrast, our SatCLE effectively balances the smooth and gradual variation of neighboring values with the semantic and geographic distinctions of different locations. This improvement is primarily due to our proposed geographic and semantic refinement strategy, allowing for a more natural capture of the continuous changes across different regions of the Earth's surface.

*4.5.3 Location Embedding Similarity.* We proceed with a qualitative analysis aimed at evaluating the degree to which various models have implicitly encoded the representations of distinct geographical locations. Specifically, we examine the similarity between location embeddings, quantified through the cosine distance between the embedding of a given location, $L$, and a reference location, $L^*$. In Figure 8, we use Beijing, China, as the reference point and visualize the resulting similarity map of the surrounding regions.

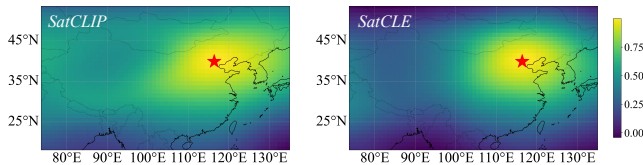

**Figure 8: Location Embedding Similarity. The location of Beijing, China is marked with the red star.**

As we can see, compared to SatCLIP, our model demonstrates more accurate location embeddings compared to SatCLIP, which can be observed in two aspects: 1) *More accurate similarity range.* In the right figure, our SatCLE model captures a more concentrated region of similarity around the reference location (*i.e.*, Beijing), underscoring its enhanced capacity to accurately represent local geographic areas. In contrast, the left figure reveals a more diffuse similarity distribution, where distant regions still display relatively high similarity. This indicate that SatCLIP may not effectively capture differences between distant locations. 2) *More effective capture of abrupt transitions.* In the right figure, similarity decreases significantly as the distance from Beijing increases, demonstrating the model's superior ability to identify abrupt changes between geographic locations. On the other hand, the left figure shows that the similarity transition is too smooth and fails to fully reflect the significant differences between positions.

## 5 Conclusion and Future Work

Location embedding has attracted widespread attention due to its characteristics of "one single unified input (*i.e.*, coordinates) serving diverse downstream tasks". In this work, we investigate the continuity issue of location embedding from both semantic and spatial perspectives and propose respective geospatial and semantic refinement strategy to bridge the gap. The state-of-the-art performance validates the effectiveness of our model. In the future, we envision the development direction can leverage knowledge distillation techniques to improve location embedding performance by utilizing the knowledge base of large models in other modalities [41, 65].

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

# Appendix

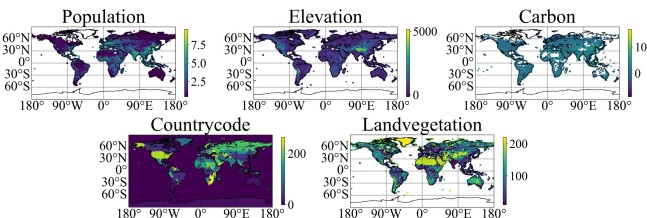

**Figure 9: Downstream Dataset Distribution.**

## A    Continuous Modeling Learning

In recent years, the concept of continuity modeling has garnered significant attention within the deep learning community, demonstrating substantial potential in fields such as time-series analysis [10, 24], visual signal processing [13, 35, 51], and drug discovery [47]. [10] introduced Neural Ordinary Differential Equations (Neural ODEs), which integrate differential equations into neural networks, enabling continuous-time reasoning that surpasses the capabilities of traditional discrete models like Recurrent Neural Networks (RNNs) networks. In the domain of computer vision, [51] advanced implicit neural representations by incorporating periodic activation functions, such as SIREN, to robustly fit complex signals. This approach has proven to enhance the performance of models in tasks like image reconstruction and neural rendering. Additionally, LIIF [13] achieves more efficient image super-resolution through localized image processing, significantly improving generalization across varying resolution inputs. Furthermore, the SSIF model [35], specifically designed for spatial-spectral super-resolution tasks, effectively combines spatial and spectral data to generate high-resolution imagery, demonstrating exceptional performance in remote sensing analysis. We introduce the concept of continuity in the context of location embedding for the first time, and decouple it into two distinct aspects, addressing each dimension with tailored approaches.

## B    Modality Representation Learning

### B.1    Visual Modality.

We leverage ViT [19] as the visual encoder to process satellite imagery. For a satellite image $I \in \mathbb{R}^{H \times W \times 3}$ with a centroid location at the specific coordinate $L$. We first split it into a sequence of 2D patches $I_P \in \mathbb{R}^{N \times (P^2 \cdot C)}$, where $C$ represents the number of channels, $p$ is the resolution of each image patch, and $N = HW/P^2$ denotes the length of patch sequences. Then we linearly embedded the visual patches into a dense vector with the latent vector size D: $e_P^I = W_P I_P^\top + b_P$, where $W_P$ and $b_P$ are learnable parameters. We also add a learnable embedding $I_{cls}$ (similar to BERT [16]'s [class] token) at the beginning of the patch embedding sequence to serve as the image representation. Additionally, we incorporate a learnable positional embedding $E_{pos} \in \mathbb{R}^{(N+1) \times d}$ to preserve positional information. We formalize this process as:

$$z_0^v = [I_{cls}; e_p^1; e_p^2; ...; e_p^N] + E_{pos}, \tag{8}$$

$z_0$ is subsequently fed into the ViT. The ViT architecture consists of alternating layers that perform the multi-head self-attention (MSA) operation [57] and fully-connected (FC) layers. Additionally, layer normalization (LN) is applied before each block, while residual connections are implemented following each block.

$$\mathbf{z'}_l^v = \text{MSA}(\text{LN}(\mathbf{z}_{l-1}^v)) + \mathbf{z}_{l-1}^v, \quad l = 1 \dots L, \tag{9}$$

$$\mathbf{z}_l^v = \text{FC}(\text{LN}(\mathbf{z'}_l^v)) + \mathbf{z'}_l^v, \quad l = 1 \dots L. \tag{10}$$

Other suitable vision encoders, such as ResNet [22] or Swin Transformer [29], can be seamlessly integrated into this framework.

### B.2    Location Modality.

Spherical harmonics have a longstanding tradition in the geosciences, where they are employed to represent various physical field theories [43, 52]. In this work, we propose to utilize spherical harmonic basis functions as positional embeddings, which could offer an effective global coverage, including coordinates at the poles.

$$\mathbf{z}^l = f(\text{SH}(\lambda_i, \theta_i)), \tag{11}$$

where $\lambda_i \in [-\pi, \pi]$ and $\theta_i \in [-\pi/2, \pi/2]$ denote the longitude and latitude as global geographic coordinates. $SH$ representes the Siren location encoder [49] that utilizes spherical harmonics basis functions. $f$ can be different neural networks such as linear layers and residual layers [31]. Following [49], we leverage SirenNet [51] to encode the posion embedding.

## C    Attention Visualization

To showcase the effectiveness of removing environmental confounding factors in SatCLE, we conducted an attention visualization presented in Figure 10. As observed, the truly semantic patches can be pinpointed by high attention values across various topographical features. Subsequently, those non-causal patches undergo a semantic smoothing process to maintain overall semantic continuity.

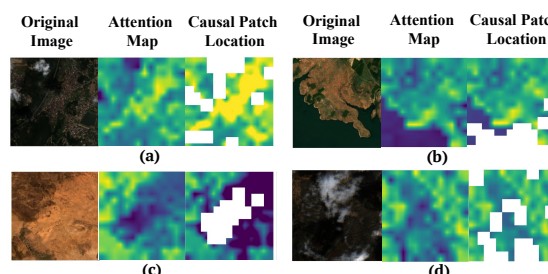

**Figure 10: Attention Visualization.**

## D    Downstream Dataset Distribution

To effectively illustrate the data distribution across downstream tasks, we visualize the downstream dataset distribution in Figure 9. As observed, all five downstream tasks achieve comprehensive global coverage.

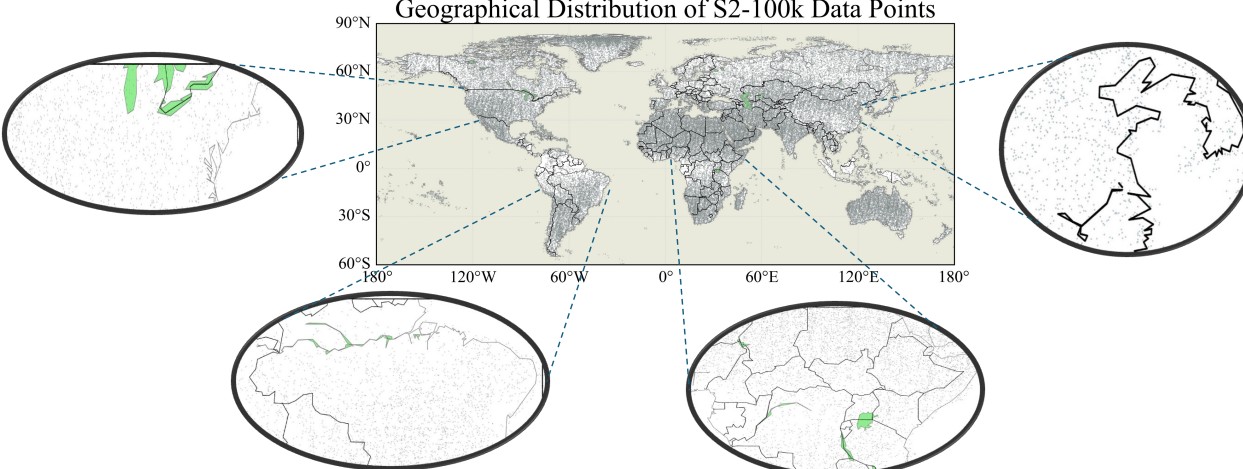

**Figure 11: S2-100k Distribution.**

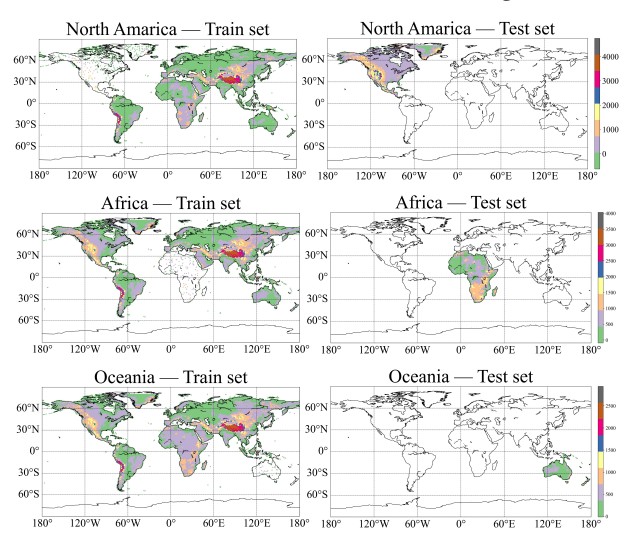

**Figure 12: Downstream Elevation Data Distribution for North America, Africa, and Oceania.**

Figure 12 demonstrates the data distribution in transferability study. We incorporate a small, uniformly randomly selected subset of data (1%) from the test continent into the training set, thereby creating a practical few-shot geographic adaptation scenario.

## E  S2-100k Distribution

To support our argument regarding spatial out-of-distribution issues, we visualize the actual distribution of the S2-100k dataset in Figure 11, focusing on specific regions that are displayed with enhanced magnification. As illustrated, the spacing between points is notably extensive. Each satellite image covers an area of 2.5 km by 2.5 km; however, the distance between adjacent points can extend to several tens of kilometers.

