# OpenReview forum: "Nature Makes No Leaps: Building Continuous Location Embeddings with Satellite Imagery from the Web"
_ACM.org/TheWebConf/2025/Conference — WWW 2025 Poster_

### Official Review · Reviewer_V9YC · 2024-11-25

**Novelty:** 6
**Technical Quality:** 6

**Review:**

This work aims to address the challenge of discontinuity in location representations by leveraging stochastic perturbation continuity expansion, graph propagation fusion, and causal theory to generate continuous geospatial embeddings from web-sourced satellite imagery. The manuscript is logically coherent, well-structured, and demonstrates a good level of methodological innovation. The experimental results are outstanding, showcasing the effectiveness of the proposed approach. However, some areas could be further improved: code availability, precise and detailed description of formulas and figures, downstream task design and so on.

**Questions:**

1. I understand that images contain rich information about geographic locations. However, beyond visual information, textual data is also crucial for understanding a location, such as historical and cultural context or people's opinions about a tourist attraction from social media. Have the authors considered incorporating textual information into their approach, and if so, how might this be used to generate more comprehensive geographic representations?
2. Will the data and code used in this work be made open-source? Alternatively, would the authors consider providing a checkpoint that can generate location embeddings?
3. In line 531, the manuscript mentions the parameters alpha and beta in the context of a formula. However, these parameters do not appear in the corresponding equations. Could the authors clarify this discrepancy?
4. What is the computational cost of the proposed method when applied to downstream tasks? This could potentially impact the practical applicability of the approach.
5. How were the downstream tasks designed? Why is it that visual signals extracted from satellite imagery can indicate information such as country codes? Additionally, would the proposed positional representation be applicable to more human-centric, location-related tasks, such as location-based recommendation systems?
6. Please provide a detailed description of Figures 4 and 7, like what is the exact meaning of the value 7.5 for SatCLIP in Figure 7?

**Reviewer Confidence:**

2: The reviewer is willing to defend the evaluation, but it is likely that the reviewer did not understand parts of the paper

**Scope:**

4: The work is relevant to the Web and to the track, and is of broad interest to the community

---

### Official Review · Reviewer_WBar · 2024-12-01

**Novelty:** 4
**Technical Quality:** 2

**Review:**

The paper develops a novel geospatial embedding vector that improves upon state-of-the-art GeoSpatial embeddings.

The paper seems to improve the existing GeoSpatial embeddings; its performance is evaluated on several tasks. The motivation behind the proposed solution is reasonable, and the experiments are relatively concluded (but not comprehensive). It is a useful and meaningful contribution; however, the presentation and arguments can be improved, and IMO, the paper is not ready for publication. There are a number of places where the statements are inaccurate or no evidence is provided to support them (see Questions), and some claims are questionable or are not properly explained (see Questions).

The key claim of this paper "specifically, to address the out-of-distribution query challenge of spatial continuity, we propose a geospatial refinement strategy comprising stochastic perturbation continuity expansion and graph propagation fusion, which transforms discrete geospatial coordinates into a continuous space" is not addressed. Learning a realistic and accurate embedding from sparse data is not a realistic task, so OoD is not achievable using the setup proposed in this work (see Questions). Additionally, it is not illustrated that the continuity problem is solved.

**Questions:**

## Definition and Solution for OoD

**“In the downstream task prediction phase, we get rid of the vision encoder and only utilize a frozen location encoder for predicting downstream tasks by simply finetuning multi-layer perceptrons with a few trainable parameters.”**
I was wondering how OoD prediction is possible for a downstream task. If a location $X_1$ and its neighbors are not seen during training (OoD), why should one expect that having the location coordinates, without images, would be sufficient for out-of-sample prediction? Especially in Figure 11, it seems training data points are sparse, and making a prediction only based on the coordinate values (not images) for a random location is an unrealistic task!

**“During inference, the input data can correspond to any arbitrary location on Earth, resulting in an out-of-distribution spatial shift, i.e., 𝑃(𝑋_{te}) ≠ 𝑃(𝑋_{tr}).”**
This is not correct. $X_{te} \neq X_{tr}$ does not imply $P(X_{te}) \neq P(X_{tr})$. When we draw samples from $P(X)$, and $X$ is continuous, it is unlikely that we find any two samples such that $X_i = X_j$ if $i \neq j$.

---

## Adding Noise to a Uniform Grid

It is claimed that adding noise to the location of the training sample (instead of using a uniform, non-overlapping grid) solves the continuity problem. However, it is not explicitly shown that the continuity problem is eliminated, and no evidence is provided to support the claim that the proposed method is effective. The proposed method improves performance, but improvement in performance does not imply that the continuity problem is solved.

**Section 3.2.3:**
I am not convinced that the discussion there is necessary, adds a new insight, or is even accurate. Let's suppose a 1-D setting, or even accurate: a uniform grid can be a realization of the uniform distribution. Adding noise to a uniform grid (in one dimension with periodic boundary conditions) would still result in a uniform distribution, regardless of the noise level. There is nothing special about adding noise.

**Line 564:**
“The S2-100k dataset encompasses imagery from 2021 to 2023, formatted into 256x256 pixel image patches in GeoTIFF format, with each pixel representing a resolution of 10 meters. Each satellite image is associated with the corresponding latitude and longitude of the patch’s center point, utilizing the EPSG:4326 coordinate system.”
I was wondering how the noise was added to the location here. This does not describe how the proposed method is related to this data set. Figure 11 seems pretty random and non-uniform. I am not sure how the continuity problem is addressed with this specific data set.

---

## Causal Framework

**Line 452:**
“In the context of satellite image inputs, $V$ and $E$ are complementary and mutually exclusive.”
Why are $E$ and $V$ mutually exclusive and complementary? Suppose $E$ represents pollution levels. Why should we expect it to be complementary to visual information (e.g., land vs. terrestrial, city vs. suburb)? Also, the example in Figure 3 does not make sense, as the "Oceanic region" is not a type of environmental semantic representation.

I am completely lost in the description of the implementation. Many concepts are introduced (e.g., student vs. teacher network) without proper introduction or description. Furthermore, I do not see how the implementation relates to the Backdoor adjustment.

---

## Experiments

- **Is h = {Population, Elevation, Carbon Emissions, Country Code, and Land Vegetation}?**
- **Why are no error bars provided for the measures in Table 1?**
- “The S2-100k dataset encompasses imagery from 2021 to 2023, formatted into 256x256 pixel image patches in GeoTIFF format, with each pixel representing a resolution of 10 meters.”
  If the resolution is $20 \times 10\ m^2$, why do all the maps presented in the evaluation have low resolution? If the embedding is made for an image of size $2560 \times 2560\ m^2$, how are the similarities in Figure 8 computed? The map in Figure 8 has a much lower resolution than $2.5 \times 2.5$ km$^2$.
- Also, it has not been illustrated how well the model performs when $Y$ is different from $h$.

---

## Minor Comments:

- **Line 192:**
  What does the index $i$ correspond to? Why are other variables not indexed?
- **Line 197:**
  What does "3" correspond to? The number of colors? In the experiments, it is mentioned that the dataset consists of 12 channels. Am I missing something here?
- **Line 205:**
  The environmental or socio-economic conditions are a function of time and smoothing scale. How are these values determined? At which time? If averaging has been performed, what kind of averaging or smoothing is used?
- **Line 207:**
  $L$ needs to be defined before being used.
- **Figure 2:**
  The use of notation $L$ and $I$ seems inconsistent with Definitions 1, 2, and 3.
- **Line 218:**
  “applications. [8, 33, 38, 53, 69].” → “applications [8, 33, 38, 53, 69].”
- **Line 256:**
  "After comprehensive literature review, we summarize two main approaches in geospatial contrastive pretraining are: (a) Vision Pretraining, which uses satellite imagery to represent geographic locations, reducing the need for ground-based surveys.”
  This is not a comprehensive review. Is this an artifact of using an LLM for writing and editing? There are many places where the text is unnecessarily verbose and occasionally inaccurate or out of context.
- **Line 482:**
  “First, we can re-express $P(H | do(V))$ using Bayes’ theorem.”
  Incorrect. The law of total probability is used here, not Bayes’ theorem.
- **Lines 498–501:**
  “The overall structure is a self-distillation framework that maintains a student network (green) and a teacher network (orange), both sharing the same architecture but having different parameters.”
  This is the first time the concepts of self-distillation, student networks, and teacher networks are mentioned. These concepts should have been introduced and discussed earlier.
- **Line 682:**
  The hyperlinks for citations 30 and 82 do not work.
- **Figure 4:**
   A log scale for population density might be more suitable.

**Reviewer Confidence:**

3: The reviewer is confident but not certain that the evaluation is correct

**Scope:**

3: The work is somewhat relevant to the Web and to the track, and is of narrow interest to a sub-community

---

### Official Review · Reviewer_yEcB · 2024-12-02

**Novelty:** 6
**Technical Quality:** 5

**Review:**

## Summary

This work focuses on continuous location embedding in satellite imagery. Authors found two challenges in continuity including spatial continuity and semantic continuity that existing methods did not solve: (1) the discrete satellite imagery samples lead to out-of-distribution (OoD) query problems; (2) the environmental confounders in satellite imagery lead to semantic discontinuity. They propose a SatCLE framework comprising geospatial refinement strategy and causal refinement strategy to address and mitigate two challenges. The experiments compare to existing state-of-the-art works and show outperforming performance.

## Pros

1. This paper is well-written. It clearly explains the motivation and states the problem.
2. The The paper addresses two key problems with current location embedding methods: (i) Spatial Continuity: Existing methods struggle to accurately represent locations that were not present in the training data. This is due to the sparse sampling of satellite images, which results in large gaps between the locations used for training. (ii) Semantic Continuity: Even when locations are close, environmental confounders in satellite images (e.g., crops surrounding a castle) can cause models to incorrectly represent their semantic similarity. This leads to semantically similar locations being mapped to distant points in the embedding space.
3. It proposes interesting approaches to address the above problems.
 (i) Geospatial Refinement, which aims to enhance spatial continuity by transforming discrete geographic coordinates into a continuous space. It achieves this through a two-step process: Stochastic Perturbation Continuity Expansion (SPCE) which adds random perturbations to the original coordinates, effectively filling the gaps between sampled locations.  Graph Propagation Fusion which employs a GNN to integrate features of the perturbed locations, ensuring a smooth and continuous representation in the latent space. (ii) Causal Refinement which addresses semantic continuity by mitigating the influence of environmental confounders. It leverages causal inference theory and involves (a) Causal Patch Localization, identifying semantically meaningful patches in the satellite image using attention mechanisms, and (b)  Attentional Semantic Smoothing, reducing the impact of confounding factors by replacing irrelevant patches with information from nearby causal patches.
4. The main novelty is that its proposed SatCLE framework, which combines spatial and semantic continuity, generates more accurate and robust location embeddings. It shows this by conducting experiments in a number of downstream applications, namely population density prediction, elevation estimation, and land vegetation classification.


## Area can be improved

1. In line 151 to 154, “To address the first challenge, …”, line 159, “To address the second challenge, …”, It’s unclear what the first/second challenge is.  It would recommend that add a tag(e.g., challenge 1) to line 123 “leads to out-of-distribution (OoD) query problems” and line 135 “lead to semantic discontinuity” for clear representation.
2. Experimental visualizations (Figure 4, Figure 7) lack instructions that guide readers to understand the superior performance of SatCLE. What do the circled parts mean in Figure 4? What does color change mean in Figure 7?
3. Data noisy: It would be good to demonstrate whether the performance of SatCLE is affected by the quality of satellite imagery, such as low-resolution, outdated satellite imagery.
4. Ablation study: The causal refinement strategy relies on the accuracy of identifying causal patches. It would be good to demonstrate the effect of the accuracy of identifying causal patches on the performance of SatCLE.

**Questions:**

See Area can be improved in Review

**Reviewer Confidence:**

4: The reviewer is certain that the evaluation is correct and very familiar with the relevant literature

**Scope:**

4: The work is relevant to the Web and to the track, and is of broad interest to the community